# Phase Formation and the Electrical Properties of YSZ/rGO Composite Ceramics Sintered Using Silicon Carbide Powder Bed

Artem Glukharev [1,2,*], Oleg Glumov [1], Ivan Smirnov [1], Evgeniy Boltynjuk [1,3], Olga Kurapova [1] and Vladimir Konakov [2]

[1] Saint Petersburg State University, Institute of Chemistry, Universitetskya nab 7/9, 199034 St. Petersburg, Russia; glumov@yahoo.com (O.G.); i.v.smirnov@spbu.ru (I.S.); boltynjuk@gmail.com (E.B.); olga.yu.kurapova@gmail.com (O.K.)

[2] Institute for Problems of Mechanical Engineering of the Russian Academy of Sciences, V.O. Bolshoj pr., 61, 199178 St. Petersburg, Russia; vgkonakov@yandex.ru

[3] Institute of Nanotechnology, Karlsruhe Institute of Technology, Hermann-von-Helmholtz-Platz 1, 76344 Eggenstein-Leopoldshafen, Germany

[*] Correspondence: a.glukharev@spbu.ru

**Abstract:** Fully stabilized zirconia/graphene composites are very promising advanced structural materials having mixed ion–electron conductivity for energy storage and energy conversion applications. The existing methods of the composite manufacturing have a number of disadvantages that limit their practical use. Thus, the search for new sintering methods is an actively developing area. In this work, we report for the first time the application of the SiC powder bed sintering technique for fully stabilized zirconia (YSZ) composite fabrication. The reduced graphene oxide (rGO) was used as a graphene derivative. As a result, well-formed ceramics with high density and crystallinity, the maximal microhardness of 13 GPa and the values of the ionic conductivity up to $10^{-2}$ S/cm at 650 °C was obtained. The effects of the sintering conditions and rGO concentration on the microstructure and conductivities of ceramics are discussed in detail. The suggested powder bed sintering technique in a layered graphite/SiC/graphite powder bed allowed well-formed dense YSZ/rGO ceramics fabrication and can become a suitable alternative to existing methods for various oxide ceramic matrix composite fabrication: both conventional sintering and non-equilibrium (SPS, flash sintering) approaches.

**Keywords:** ceramic matrix composites; yttria-stabilized zirconia; graphene; powder bed sintering; impedance spectroscopy

## 1. Introduction

For the past decade, ceramic matrix composites (CMCs) modified with graphene or its derivatives have been attracting great attention as a novel class of materials for new energy storage devices and batteries [1–3], solid oxide fuel cells [4] and thermal barrier coatings [5,6] with improved hardness, fracture toughness, bending strength and wear resistance [7]. Graphene additive limits grain growth in the ceramic matrix during sintering resulting in the composite reinforcement. Stabilized-zirconia-based composites are of particular interest since the introduction of graphene to the zirconia matrix results in mixed ionic-electronic conductivity [8–10]. The sintering step is a key factor to retain graphene additive in the zirconia matrix and to achieve the desired properties of CMCs. The fabrication of stabilized zirconia/graphene composites requires high-temperature sintering (1300–1600 °C), long exposure times and a vacuum or inert atmosphere [11–13]. To date, a number of zirconia/graphene composite sintering techniques have been proposed in the literature [7,14]. They can be divided into two groups: equilibrium (convectional) sintering and non-equilibrium (field-assisted) sintering [15].

The first group of methods includes hot uniaxial pressing, hot isostatic pressing, pressureless sintering and some others [16,17]. The sintering is realized through heat convection from the heating element to the sample, and it takes several hours to archive the full densification of a composite. The main advantages of the listed methods are (i) the possibility to reach a high degree of crystallinity; (ii) a wide range of starting oxide and non-oxide powder materials that can be sintered; and (iii) the dimensions and shape of the samples are variable and are limited only by furnace chamber size. However, the family of graphene materials is very sensitive to oxidation in the presence of residual oxygen up to $10^{-3}$ atm at elevated temperatures (>300 °C) even when they are incorporated into a metal or ceramic matrix [10]. For that reason, sintering is usually conducted in a deep vacuum or in an argon atmosphere. However, successful sintering under these conditions requires a significant complication of the experimental setup (high-power vacuum pumps, oxygen control sensors, etc.), which increases the cost and imposes restrictions on the number of simultaneously sintered samples. In [8,9], fully yttria-stabilized zirconia (YSZ)/reduced graphene oxide (rGO) composites were sintered using both conventional sintering in air and sintering in a deep vacuum. It was shown that the rGO additive acts as a grain growth inhibitor, which is burned out completely during annealing in air. As a result, the refined microstructure of ceramics and coinciding temperature dependencies of grain and grain boundary conductivities in the whole temperature range were achieved. Vacuum annealing resulted in YSZ/rGO composites with high ionic conductivity and high crystallinity; however, the absence of electron conductivity was shown.

Non-equilibrium sintering approaches including spark plasma sintering (SPS) are most commonly applied to obtain stabilized zirconia/graphene composites with desired mechanical and electrical properties [8,10,18]. This group of sintering techniques is based on the direct energy transfer from a source to the sample via an electric current or electromagnetic radiation. A high rate of temperature increase up to 1000 °C/min leads to the shortening of the sintering time to several minutes; this time, shortening limits the grain growth, allows sintering of the fully dense specimens and preserves graphene from oxidation. As it was shown in [8], SPS allows one to retain graphene in the zirconia matrix and leads to the mixed electronic-ionic conductivity of the composites. However, a cycle heating experiment showed that the continuous graphene framework is destroyed by the grain growth which takes place because of an incomplete crystal structure formation during the sintering process. The SPS process cannot be used for a batch production of zirconia/graphene composites because of the equipment complexity and limited geometry of specimens. Thus, there is a need for new simple and cost-effective fabrication routes for ceramic/graphene composites, and especially zirconia/graphene composites, where a deep vacuum and/or special equipment are not necessary. For instance, a fast room temperature and air fabrication of 3YSZ/rGO composites by flash sintering (FS) has been recently reported in [14]. Dense composites with a homogeneous distribution of rGO in the zirconia matrix, enhanced electrical conductivity and mechanical properties compared to pure 3YSZ were obtained.

The powder bed sintering approach is a promising alternative to overcome the existing disadvantages of listed equilibrium and non-equilibrium sintering approaches. The essence of this approach is heating the sample fully covered with the mother powder of the same composition or some inert powder such as SiC. In this case, the direct contact with the furnace chamber atmosphere is eliminated. A powder bed can perform various functions: limiting diffusion of the material's components from the sample bulk, protecting sensitive components from oxidation, etc. The described approach prevents evaporation and oxidation of the ceramics and composite components and allows respective reactions in the sample bulk to occur. In works [19,20], $Si_3N_4$-based ceramics were successfully sintered at 1650–1800 °C under a nitrogen atmosphere. In [19], samples were placed into a graphite container with a $Si_3N_4$-BN powder bed of 1:1 ratio. The authors reported that the use of a nitride powder bed allows successful sample sintering at a nitrogen pressure of 1 atm with minimal weight loss, limited thermal decomposition of $Si_3N_4$ during heating

and high density >95% compared to for the sample sintered under the same inert atmosphere but without a powder bed. The positive effect of the application of the powder bed technique was also proven for SiC ceramics [21]. It was shown that the mother powder bed serves as an additional source of the species. Several attempts were also made to apply powder bed sintering to solid electrolyte production. For instance, perovskite-type $Li_{3/8}Sr_{7/16}Ta_{3/4}Zr_{1/4}O_3$ was successfully sintered in a mother powder bed at 1300 °C for 15 h in an alumina crucible [22]. The technique was shown to be effective to densify the pellet and reduce impurity phases of $SrTa_2O_6$ and $Sr_2Zr_2O_7$ and lithium losses. As a result, an about three-time increase in the ionic conductivity was observed compared to conventionally sintered ceramics. The use of the powder bed for zirconia/graphene composites may protect graphene additive from oxidation during sintering and hence could allow one to combine the advantages of equilibrium and non-equilibrium sintering approaches: high crystallinity, fine microstructure, simple technical realization and elimination of such limitations as sample geometry or the necessary minimum level of the material conductivity. Thus, the goal of the present work was to develop a suitable powder bed sintering method for the ceramics/graphene composite production, determine the relationship between the amount of introduced graphene and phase formation, microstructure, conductivity and mechanical characteristics and investigate the powder bed effect on the obtained properties.

## 2. Materials and Methods

### 2.1. YSZ-rGO Composite Manufacturing

Composites were obtained from the individual powders of 8 mol.% yttria-stabilized zirconia (YSZ) and reduced graphene oxide (rGO). YSZ precursor powder was synthesized by a sol–gel method in a variant of reversed co-precipitation. Precipitation of the 0.1 M aqueous solution of mixed nitrate hydrates $Y(NO_3)_3 \cdot 6H_2O$ (Acros Organics, Geel, Belgium, 99.9%) and $ZrO(NO_3)_2 \cdot 6H_2O$ (Acros Organics, Geel, Belgium, 99.5%) was realized by dropwise addition of 1 M ammonia solution (LenReactiv Ltd., St Petersburg, Russia, c.p.). The obtained gel was washed to a neutral pH then freeze-dried (Labconco, 1 L chamber, Kansas City, MO, USA; 293 K, 24 h, P 0.018 mm Hg) and annealed at 700 °C for 3 h (to start the formation of cubic solid solution) and then milled in a planetary ball mill (Pulverisette 6, Fritsch, Germany, 420 rpm, 24 reverse cycles of 5 min each) to diminish powder agglomerate size. A detailed description of used synthesis procedures is provided elsewhere [9].

Oxidation/reduction route was chosen for the reduced graphene oxide (rGO) production. Oxidation from the commercial highly oriented pyrolytic graphite (Active-nano Ltd., Russia) was performed via Hummers technique [23]. The obtained graphite oxide was washed by distilled water until the neutral pH of filtrate, treated in an ultrasonic bath for 30 min (bath Sapphire 2.3 TTC, Russia, frequency 35 kHz, 0.1 kW power) for the delamination of individual graphene oxide layers then dried at 100 °C for 3 h and then thermally reduced in two steps: exposure at 300 °C for 1 h with the following microwave irradiation treatment (household microwave oven, 2.45 GHz, 750 W, 30 s) in the argon atmosphere. Additional microwave irradiation step allows one to increase the quality of the graphene basal plane and to decrease the number of defects [24]. Thus, the individual components were obtained.

YSZ and rGO powders were mixed in 4 different ratios (see Table 1) and milled in the planetary ball mill to prepare powder composites, break particle agglomerates and reach homogeneous distribution of the carbon additive in the ceramic matrix. The optimal amount of the graphene additive providing the formation of a "core-shell" type composite with YSZ agglomerate covered with the single rGO layer was calculated using simple mathematical model proposed in recent papers [8,9].

**Table 1.** Sintered samples: powder compositions and sample designation.

| Sample | Brutto-Composition | Sintering Conditions |
|---|---|---|
| Z0 | 91ZrO$_2$-8Y$_2$O$_3$ (8YSZ, mol.%) | 1550 °C; |
| Z0.25 | YSZ + 0.25 wt.% rGO | 3 h; |
| Z1 | YSZ + 1 wt.% rGO | 3-layer graphite/SiC/graphite |
| Z2 | YSZ + 2 wt.% rGO | powder bed |

Composite powders were compacted into pellets with the diameter of ~8 mm and thickness of ~5 mm by cold uniaxial pressing with a force of 49 kN (Mega kck-50A, 50 t, Spain) and exposure time of 5 min and then additionally compressed by cold isostatic pressure with a force of 152 MPa. The obtained green body samples were placed into a three-layer powder bed (Figure 1) and sintered at 1550 °C for 3 h in air. Sintering conditions (time and temperature) were chosen according to recent work of authors [9], where the YSZ-rGO samples were sintered without the use of a powder bed. Silicon carbide was chosen as a main component of the powder bed because of its inertness against YSZ and ability to bind oxygen with the formation of SiO$_2$ continuous layer preventing further O$_2$ diffusion and oxidation process [25]. Top graphite layer was added to bind oxygen at low temperatures, where SiC remains relatively inert. Bottom graphite layer was added to preserve carbon concentration excess and limit its diffusion out of specimens.

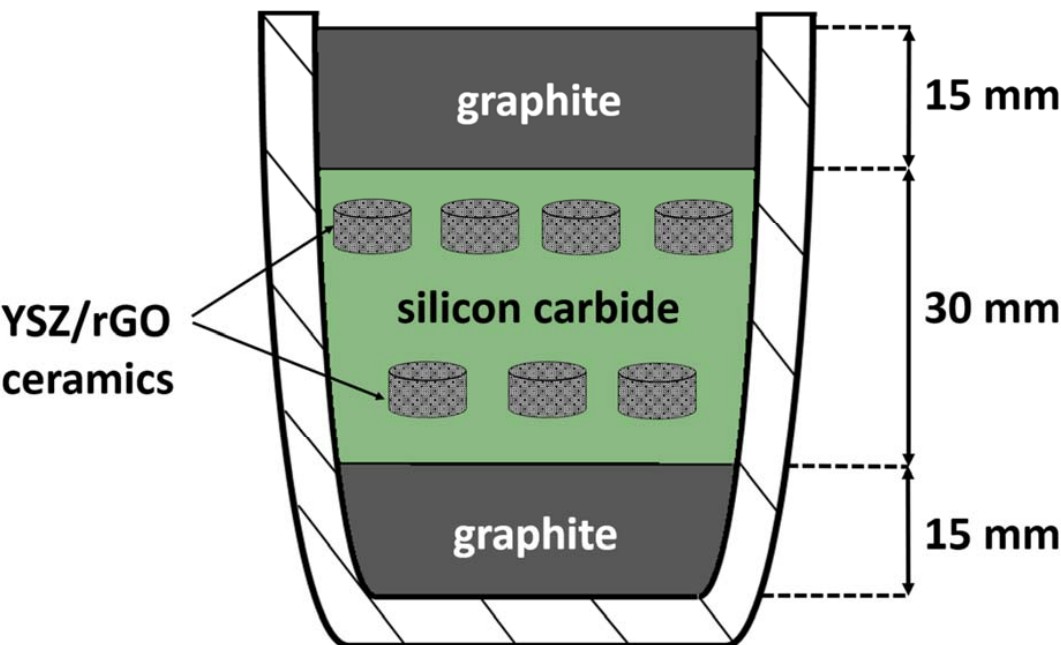

**Figure 1.** Schematic cross-section image of the alumina crucible with sintered ceramics in 3-layer graphite/SiC/graphite powder bed. Ten specimens are placed in two lines.

*2.2. Characterization of the Obtained YSZ-rGO Ceramics*

X-ray diffraction method (SHIMADZU XRD-6000, Cu-K$_\alpha$ = 1.5406 Å) was used for the phase composition analysis. Phase identification was carried out via Powder Diffraction File database (PDF-2, 2021) [26]. The ratio of crystalline and amorphous phases was calculated based on obtained XRD data using the standard diffractometer software. Crystallite sizes were estimated via Scherrer equation:

$$d_{cryst} = \frac{K\lambda}{\beta \cos \theta} \tag{1}$$

where $d_{cryst}$—is the crystallite size in nm, $K$—Scherrer constant (~0.9 for spherical particles), $\lambda$—X-ray wavelength, $\beta$—line broadening at half of the maximum intensity, and $\theta$—Bragg angle.

Raman spectroscopy (SENTERRA (Bruker), laser irradiation wavelength 532 nm and 785 nm, irradiation power of 2 mW and 10 mW, respectively) was used for the detailed investigation of the rGO state. Densities of the sintered specimens were measured using the Archimedes method in isopropyl alcohol (scales RADWAG 220 c/xc with a density measurement kit, Poland). The theoretical density was calculated using the rule of mixture with the density of pure YSZ equal to 5.96 g/cm$^3$ (82–1244. JCPDS-International Centre for Diffraction Data) and the density of rGO equated to graphite density 2.26 g/cm$^3$ [27], respectively. Obtained ceramic samples were subjected to microstructure investigations with the following surface preparation procedure: filling of the specimen with an epoxy resin compound, aging until the complete resin solidification and polishing by the series of abrasives up to 1 μm polycrystalline diamond suspension (MetaDi Supreme, Buehler, Lake Bluff, Illinois, USA). No additional etching was used. Prepared samples of each composition were studied by high-resolution scanning electron microscopy (Zeiss Merlin with GEMINI-II optical system and accelerating voltage 21 kV). For elemental mapping, X-ray energy-dispersive spectroscopy (microanalysis spectrometer Oxford Instruments INCAx-act X-ray) was used for elemental mapping. Microhardness of sintered ceramics was measured via micro Vickers hardness method (Tester Shimadzu HMV-G21DT with CCD camera microscope) with applied force of 2.9 N (HV0.3) 50 times for each specimen. Surfaces of the samples were prepared by the same polishing procedure as for microstructure investigations.

### 2.3. Electrochemical Study

The conductivity of the YSZ-rGO ceramics at different temperatures was studied via electrochemical impedance spectroscopy method (Potentiostat/Galvanostat Autolab PGSTAT 302 N). A central sector with width of ~2.5 mm was cut from a sample of each composition for the investigation. Measurements were conducted in N$_2$ atmosphere with residual oxygen pressure less than $10^{-3}$ atm with silver paste electrodes; the data for each 50 °C in a range from room temperature to 700 °C were taken. Impedance spectra were analyzed using Autolab NOVA and Impfco software. Activation energies of the conductivity were estimated using Arrhenius equation from linear temperature regressions of the conductivity:

$$\sigma = \sigma_0 \exp\left(\frac{E_a}{RT}\right) \qquad (2)$$

where $\sigma$ and $\sigma_0$ are specific conductivities, $E_a$ is activation energy of conductivity, $R$—universal gas constant, $T$ is the temperature.

### 3. Results

### 3.1. Phase Composition

The phase composition of the initial powder mixtures was studied in the first step of the work. The formation of the cubic YSZ solid solution with the crystallinity values of ~30% was confirmed for all studied compositions with the graphene contents of 0.25–2.5 wt.%. At the same time, any reflexes from rGO or carbon were absent in the XRD patterns of composite powders, indicating a high degree of homogeneity and the uniform carbon additive distribution in the ceramic matrix. Raman spectra analysis confirmed these assumptions and showed the formation of "core-shell" type composites. The optical photographs of the sintered Z0–Z2 samples are shown in Figure 2.

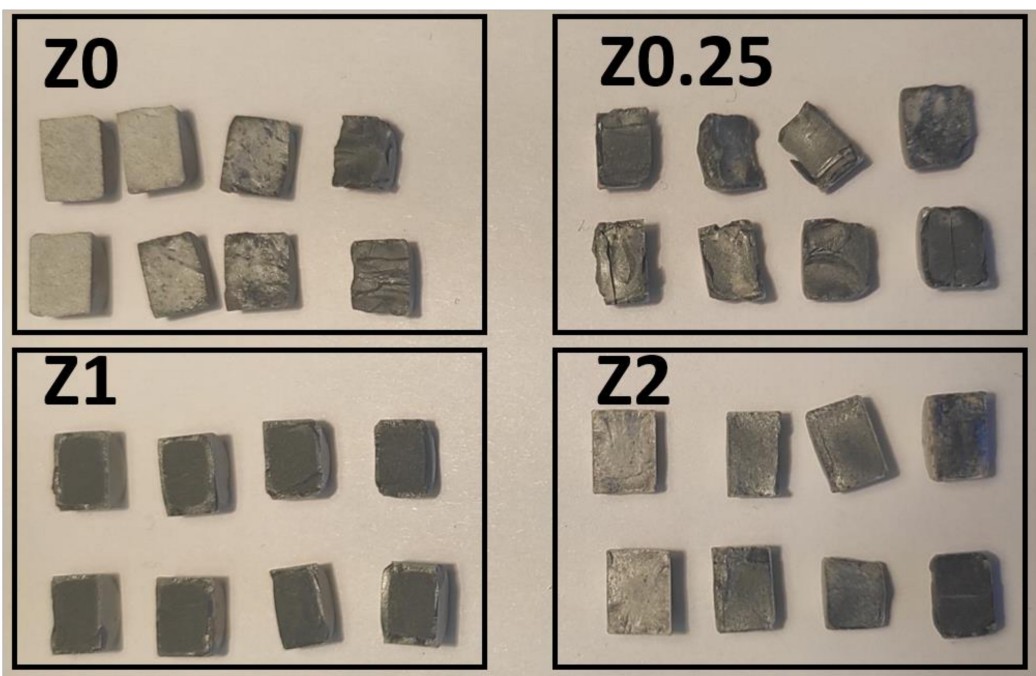

**Figure 2.** Optical photographs of the sintered Z0–Z2 samples.

The XRD patterns of sintered ceramic materials are presented in Figure 3. All peaks correspond to the zirconia-based cubic solid solution with 2θ at 29.5, 34, 51, 59, 62.5 and 73.5°, respectively. Peaks that can be related to GO at 2θ ~12° or rGO at 2θ ~25° are absent [28]. However, the main graphite peak at ~26.5° is slightly seen in the X-ray patterns of all the samples with the intensity close to the signal of the baseline.

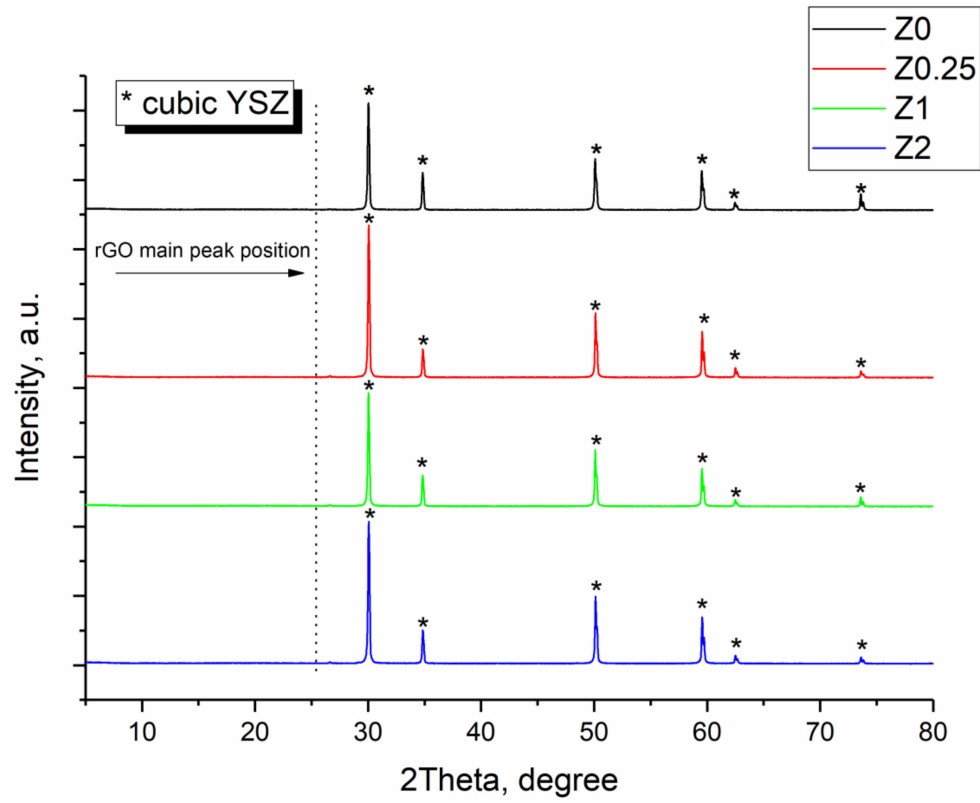

**Figure 3.** XRD patterns of the obtained samples. The symbol "*" refers to cubic YSZ phase.

Crystallinity (K, %) and the crystallite size ($d_{cryst}$, nm) of the Z0–Z2 samples sintered from pure YSZ (Z0) and composites (Z0.25–Z2) are presented in Table 2. Z0 ceramics has the highest value of K ≈ 98%, being close to the theoretical one. The data shown in Table 2 prove the completeness of the YSZ crystalline phase formation. With the addition of 0.25 wt.% of rGO, the crystallinity slightly decreases to ~94%, being in accordance with the previous studies [8,9]. The (further) increase of the carbon content does not lead to the further decrease of crystallinity. Crystallite sizes for the Z0–Z2 samples are about 110–120 nm, which is higher than the value for YSZ ceramics sintered at 1150 °C for 3 h in air (35 nm) [29].

**Table 2.** Crystallinity, crystallite size and relative density of the sintered materials.

| Sample | Z0 | Z0.25 | Z1 | Z2 |
|---|---|---|---|---|
| K, % | 98 | 94 | 93 | 94 |
| $d_{cryst}$, nm | 115 | 112 | 114 | 118 |
| Relative density, % | 94.5 ± 1.3 | 96.3 ± 1.3 | 97.1 ± 0.4 | 95.6 ± 0.7 |

A low signal from the carbon phase in the ceramics may indicate either its homogeneous distribution (as it was observed for powder composites in [9,30]) or rGO evaporation/burnout during the sintering process. For more accurate study of the carbon state in the samples, Raman spectroscopy was used. The spectrum of pure microwave-reduced rGO was obtained previously in [8]; it corresponds to some layered graphene having a decreased number of defects compared to the thermally reduced graphene oxide. Raman spectra of sintered Z0–Z2 ceramics are presented in Figure 4. A typical spectrum can be divided into two parts: an area from 0 to 1000 cm$^{-1}$, which is referred to as YSZ with characteristic bands at ~150, ~280 and 600 cm$^{-1}$ [31], and an area of rGO from 1000 cm$^{-1}$ to 3000 cm$^{-1}$ with 1350 (D), 1550 (G) and 2750 (2D) cm$^{-1}$ bands. However, a number of very intensive bands at the low shifts from 0 to ~1250 cm$^{-1}$ (observed at an excitation wavelength of both 532 nm and 785 nm) were registered for all samples studied (Figure 4a). In particular, they are: a band at 970 cm$^{-1}$ and a band at 480 cm$^{-1}$ with a shoulder at 320 cm$^{-1}$. All of them could be hardly attributed to the YSZ phase. The most intense broad band can be attributed to the transverse optic phonon of the amorphous Si and the second most intense peak to SiC [32].

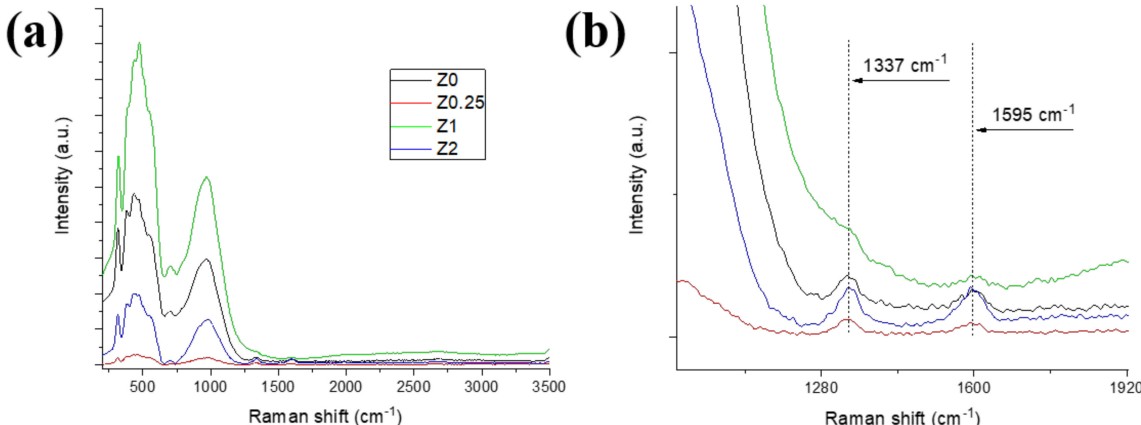

**Figure 4.** (**a**) Raman spectra of the sintered YSZ-rGO composite ceramics, (**b**) detailed rGO area spectra.

The magnified areas of spectra at 1000–2000 cm$^{-1}$ corresponding to rGO main bands are presented in Figure 4b. Both *D* and *G* bands at ~1340 cm$^{-1}$ and ~1600 cm$^{-1}$ are distinguished for all samples. The intensities of the bands are comparable to the background level, which indicates that the carbon content in the sintered samples is low.

### 3.2. Microstructure and Density of Sintered Ceramics

The experimental density data are presented in Table 2. The relative density of Z0 ceramics of 94.5 ± 1.3% was obtained using the crystallographic data for cubic YSZ. The addition of 0.25 wt.% rGO to the ceramic matrix causes the density increase up to 96.3 ± 1.3% with the same standard deviation. A further increase up to 1 wt.% rGO results in the highest density and minimal standard deviation of 97.1 ± 0.4% in the entire series of ceramics. For the Z2 sample, with 2 wt.% rGO, a slight decrease of density to 95.6 ± 0.7% takes place. Thus, a minimum of defects and the most uniform structure are expected for Z1 ceramics. SEM images of Z0–Z2 ceramics are presented in Figures 5–7. Structures of surfaces (Figures 5 and 6) can be divided into the near-surface layer and the bulk structures. All the near-surface layers contain an extensive microcrack network (Figure 5). Along with that, they contain a number of pores, and their amount increases with the increase of rGO content, which may be due to carbon additive agglomeration in the zirconia matrix during sintering.

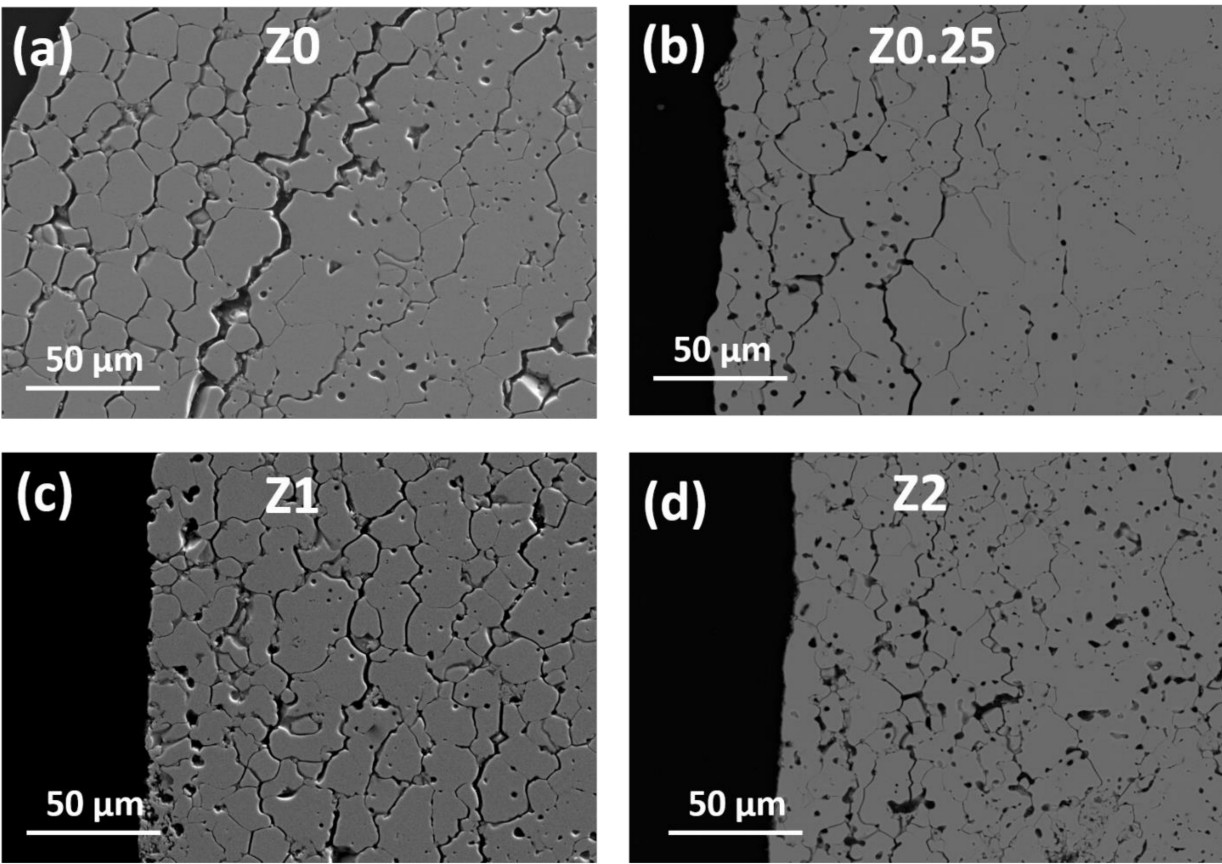

**Figure 5.** SEM images of the near-edge surfaces of sintered samples: (**a**) ceramics Z0, (**b**) ceramics Z0.25, (**c**) ceramics Z1, (**d**) ceramics Z2.

The microstructure of the surface layer differs significantly from the more uniform ceramic bulk (Figure 6). The surface layer of Z0 ceramics contains a number of prolongated cracks (Figure 6a), and their amount significantly decreases with rGO addition up to 1 wt.% (Figure 6b,c). According to [7], the fracture toughness of composites increases due to the addition of rGO, which stops the propagation of the cracks in the ceramics matrix. For the Z2 sample, the near-surface structure gradually turns into the bulk structure (Figure 6d). Clear borders between the edge area and the bulk are visible for the Z0 and Z0.25 samples and, especially, the Z1 sample (Figure 6c).

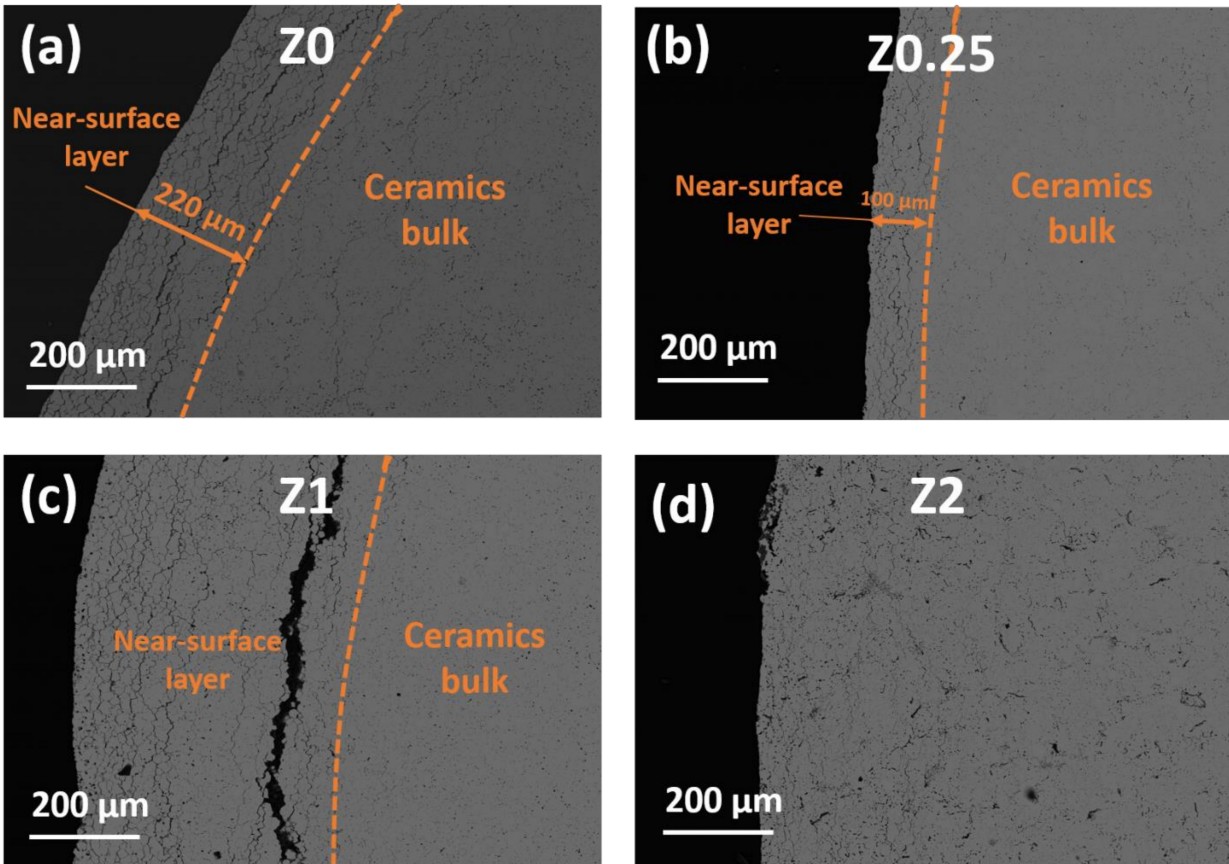

**Figure 6.** SEM images of the surface of sintered (**a**) Z0; (**b**) Z0.25; (**c**) Z1; (**d**) Z2 ceramics.

Cross sections of the specimens are characterized by different microstructures compared to surfaces (Figure 7). The image of Z0 has two structural zones. The near-edge layer is characterized by a large network of voids (Figure 7a). Furthermore, the area corresponding to the bulk of the Z0 sample has almost zero pores or defects. With the addition of 0.25 wt.% of rGO, the microstructure becomes almost monolithic and more uniform. The network of large voids vanishes, although relatively small pores are still present (Figure 7c). The boundaries of individual grains are still visible though less clearly (Figure 7d). With the increase of rGO content to 1 wt.%, the number of pores and their sizes decrease noticeably (Figure 7e–f). Along with that, grains become almost indistinguishable (Figure 7f). A further increase of rGO content to 2 wt.% leads to the formation of the extensive network of elongated pores with several large ones reaching the estimated sizes of $30 \times 20$ μm.

The performed EDX analysis shows that carbon content in the pores is ~75 at.% while it is absent on the surfaces for all studied compositions. Silicon is also absent both on the surface and inside the pores of ceramics (see Figures S2 and S4 and Tables S1 and S2 in Supplementary Materials).

*3.3. Electrochemical Study of Composites*

YSZ-based solid electrolyte possesses ionic conductivity with a hopping mechanism. According to Bauerle's circuit, an impedance spectrum of YSZ ceramics consists of two arcs, one is located at high frequencies and refers to grain resistivity while the other is located at low frequencies and refers to the resistivity of grain boundaries [33,34]. When the percolation threshold in graphene–zirconia composites is reached (at ~2 wt.%. rGO addition), the conductivity mechanism changes and the electronic component appears in the impedance spectra. In particular, two arcs disappear and total resistivity decreases for at least four orders of magnitude with the drop of activation energy from 1.1 eV to 0.1-0.2 eV [8].

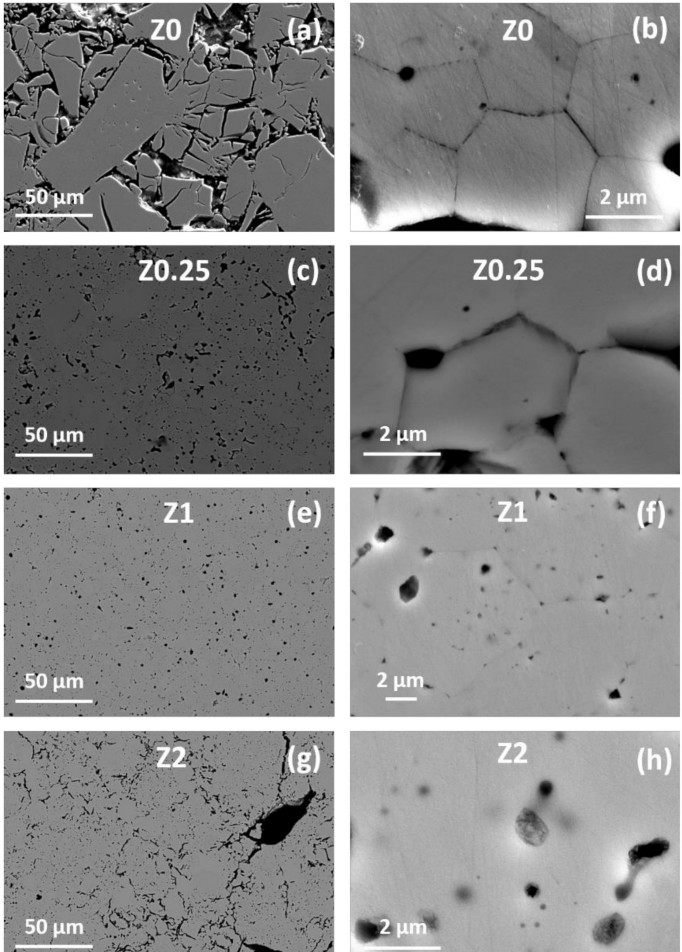

**Figure 7.** SEM images of the cross section of sintered (**a**,**b**) Z0; (**c**,**d**) Z0.25; (**e**,**f**) Z1; (**g**,**h**) Z2 ceramics.

The obtained impedance spectra of the Z0–Z2 samples are close to typical for YSZ without substantial change of conductivity, even after 2 wt.% rGO is introduced in the ceramic matrix (Figure 8). Nevertheless, the shapes of the obtained arcs are flattened, and they slightly merge with each other. The resistivities of grains (Figure 8a) for Z0, Z0.25 and Z2 ceramics are close to each other. At the same time, the resistivity of Z1 is 4.3 times higher than that for Z2 and Z0 ceramics. When the temperature increases, arcs corresponding to grain boundaries become visible (Figure 8b). A minimum value is observed for Z0, and then it monotonically increases with the rGO content increase. This results in a $R_{Z0.25}/R_{Z0}$ ratio equal to ~2.5 and $R_{Z1}/R_{Z0}$ ratio equal to ~4. The ratio for the Z2 sample is out of the existing trend with $R_{Z2}/R_{Z0}$ ~1.73.

Based on the obtained spectra, temperature dependencies of conductivity (Figure 9) were plotted using Equation (2). All the samples obey linear log Arrhenius dependence with the determination coefficient $R^2 > 0.99$. The maximum conductivity level of σ ~0.01 S/cm at 650 °C was reached for the Z0.25 sample, which is close to the typical value for YSZ ceramics [35]. The conductivities of grains for all samples are almost equal in the whole temperature range with activation energies of 1.06 and 1.10 eV, respectively. Ceramics Z0 shows relatively lower conductivity values with the same activation energy of 1.06 eV. At the same time, the Z1 sample possesses much lower conductivity and higher activation energy of 1.23 eV, which leads to the $\sigma_{Z0.25}/\sigma_{Z1}$ ratio change from ~8 at 200 °C to ~2.6 at 450 °C.

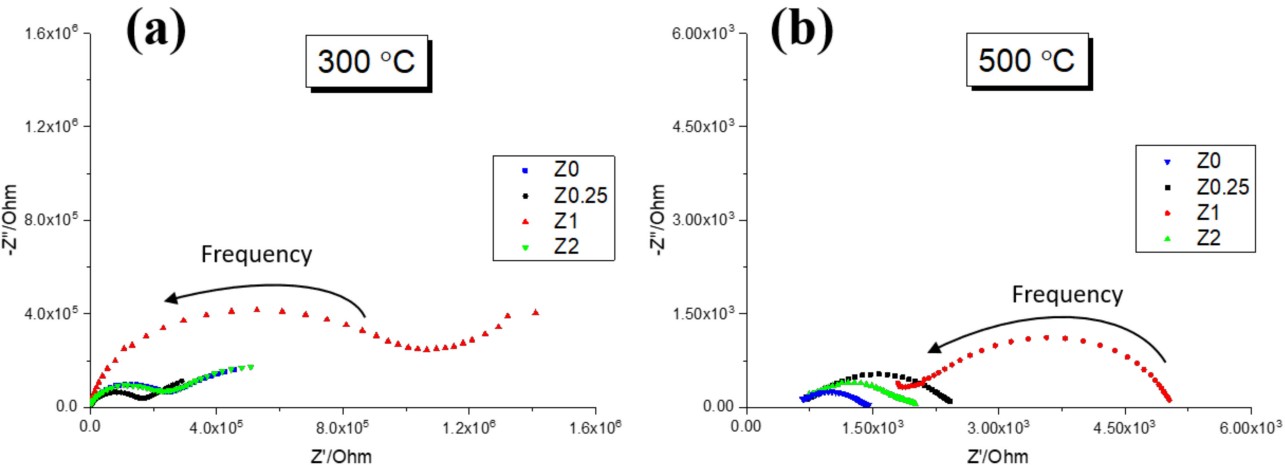

**Figure 8.** Impedance arcs of ceramics Z0–Z2 obtained at (**a**) 300 °C, (**b**) 500 °C.

**Figure 9.** Temperature dependencies of conductivity of (**a**) Z1 sample, (**b**) grain conductivity for Z0–Z2 samples, (**c**) grain boundary conductivity for Z0–Z2 samples.

Data obtained for grain boundary conductivities differ from the ones for grain components (Figure 9c; Table 3). Activation energies of grain boundaries in the series gradually decrease from 1.31 eV for the Z0 sample to 1.16 eV for the Z2 sample. It results in the following specific conductivity changes: the Z2 sample possesses the highest conductivity values of the grain boundaries at relatively low temperatures (250–350 °C); at temperatures higher than 450 °C, Z0 ceramics turns to be more conductive; the lowest conductivity among all the compositions and the highest activation energy of 1.35 eV are observed for the Z1 sample. The ratios $\sigma_{Z0.25}/\sigma_{Z1}$ are the same as seen previously, i.e., ~4.5 at 350 °C and only ~1.6 at 650 °C.

**Table 3.** Activation energy of conductivity of grain and grain boundary components of ceramics Z0–Z2.

| Sample | Activation Energy of Conductivity $E_a$, eV | |
| :---: | :---: | :---: |
| | **Grain Component** | **Grain Boundary Component** |
| Z0 | 1.10 | 1.31 |
| Z0.25 | 1.06 | 1.21 |
| Z1 | 1.23 | 1.35 |
| Z2 | 1.10 | 1.16 |

*3.4. Microhardness of Sintered Ceramics*

Microhardness data are presented in Figure 10. HV0.3 value for Z0 ceramics is $10.4 \pm 1.4$ GPa, which is close to the average values for YSZ ceramics [36]. Addition of rGO results in the microhardness increase to $12.1 \pm 1.4$ GPa for the Z0.25 sample and $13.0 \pm 1.2$ GPa for the Z1 sample, which is close to the maximal reported microhardness value for YSZ (~$15.3 \pm 0.4$ GPa [36]) and lies in the upper range for YSZ-rGO composites [37].

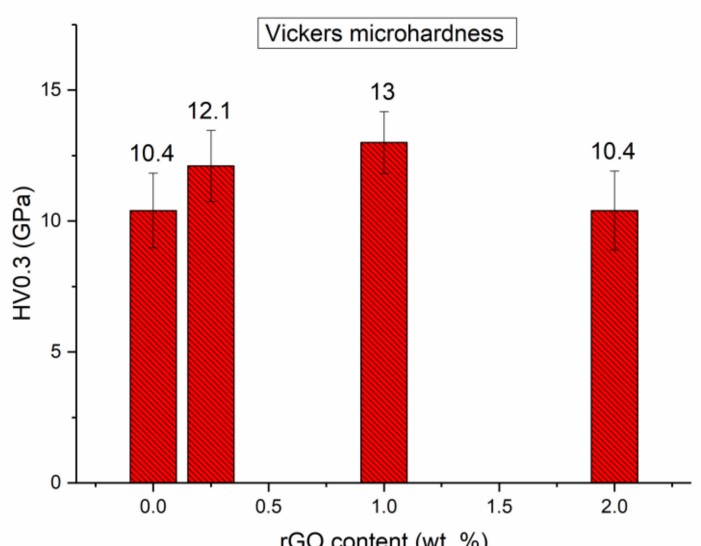

**Figure 10.** Microhardness test results for ceramics Z0–Z2.

Microhardness increases in accordance with the obtained microstructure. An addition of rGO stops the propagation of cracks in the ceramic matrix. The further increase of rGO content to 2 wt.% leads to the HV0.3 decrease back to the level of Z0 ceramics, which may indicate the agglomeration of carbon during sintering.

**4. Discussion**

As seen from the phase composition analysis and microstructure data, the chosen conditions of the powder bed synthesis, i.e., three-layer graphite/SiC/graphite bed and exposure at 1550 °C for 3 h, favor the formation of a YSZ cubic solid solution in the

samples. High relative density values of 94–97%, crystallinity values higher than 93% for all compositions and large crystallites (112–114 nm, see Table 2) prove this assumption. At the same time, the obtained Raman spectra are quite different from the typical ones for YSZ/rGO composites. Bands referring to rGO have an intensity close to the baseline, which indicates the elimination of the carbon phase after sintering (see Figure 4). The spectrum of pure Z0 ceramics with no initial rGO added shows the presence of negligible carbon bands. That points to possible carbon diffusion into a specimen from the powder bed during sintering. Intensive bands at 1–1000 cm$^{-1}$ for all compositions can be hardly attributed to YSZ. Instead, they could be referred to SiC, which also could be the source of carbon. According to [32], all crystalline silicon carbide polytypes give Raman scattering from a longitudinal optic phonon (LO) at 973 cm$^{-1}$, which is presented in spectra, and a transverse optic (TO) phonon at approximately 790 cm$^{-1}$, which is slightly seen. Another intensive broad band at ~480 cm$^{-1}$ in spectra can be referred to as amorphous SiC [38] or amorphous Si [39]. Thus, two processes taking place during sintering are expected: (i) diffusion of SiC into the specimen bulk and its decomposition with the formation of free carbon (ii) gradual rGO agglomeration at high temperatures like it was observed for SPS-sintered composites [8]. Silicon carbide powder is rather coarse, and its steady diffusion into the ceramic matrix bulk is limited (mean particle size of 36.22 μm, see Figure S5, Supplementary Materials). The comprehensive consideration of the data on microstructure, conductivity and microhardness (see Figures 6 and 8–10) allows us to conclude that they are competing with each other depending on rGO concentration.

The microstructure of the Z2 sample significantly differs from the other composites. According to SEM images (Figure 6), the structure of the near-surface layer is similar to the structure of the bulk, which points to the negligible diffusion of silicon carbide from the powder bed during sintering. At the same time, only this composition possesses an extensive network of pores with some large ones in the bulk, which, according to previous studies, corresponds to rGO agglomeration sites. Based on the EDX spectra data (see Figure S3 and Table S2 in Supplementary Materials), the large pores contain ~70 at.% of carbon, and this fact proves the above assumption. The increased values of ionic conductivity and the decrease of microhardness observed for the Z2 sample prove the assumption on rGO agglomeration. Therefore, there is less dispersed rGO observed in the intergranular space. Both the microhardness value and electrochemical behavior of the Z2 sample are close to those for pure Z0 ceramics. According to the EDX spectra and elemental distribution maps, even the bulk of Z0 ceramics contains some carbon in pores (see Figures S1, S2 and S4 and Table S1). In addition, the rather loose structure of the bulk (Figure 7a) and increased activation energy of the grain boundary conductivity obtained for the sample indicate SiC diffusion into the sample took place rather intensively. At the same time, no traces of significant agglomeration were found, and areas of the formed ceramic structure are without significant defects or large pores. Thus, it can be concluded that the diffusion process dominates for the Z0 sample which may be due to the larger carbon concentration gradient between the powder bed and the ceramic sample. From that point of view, the diffusion of SiC during sintering of Z0.25 and Z1 composites should be less pronounced. Indeed, the Z0.25 and Z1 samples have higher relative density and the most uniform structure without loosening in the bulk or large pores but have the near-surface layer. This assumption could also explain the shape and structure of the near-surface layers by the means of two-times different thermal expansion coefficients (CTE) for SiC ($\approx 4.5 \cdot 10^{-6}$ K$^{-1}$ at 873 K [40]) and for YSZ ($\approx 9.2 \cdot 10^{-6}$ K$^{-1}$ at 873 K [41]). When heated, YSZ ceramic grains have a compressive effect on diffusing silicon carbide, which, when cooled, leads to the formation of elongated pores. In addition, the uniform distribution of rGO in the intergranular space should limit grain growth and hinder oxygen vacancy transfer. Composite Z0.25 has the same type of microstructure but lower microhardness and higher conductivity, which indicates the presence of rGO in the intergranular space. The microstructure features obtained for the Z1 sample are in accordance with the decreased grain boundary conductivity and high activation energy. Finally, the Z1 sample has the

highest microhardness among all the samples sintered. The addition of 1 wt.% of rGO prevents both significant agglomeration and SiC diffusion into the bulk; i.e., it is close to the optimal concentration for powder bed sintering of sol–gel synthesized YSZ powder.

In order to reveal the nature of the carbon concentration gradient, let us consider the possible mechanisms of silicon carbide degradation during sintering from the thermodynamic point of view. In the presence of oxygen, reactions (3)–(5) are allowed:

$$\mathrm{SiC(s)} + 1/2\mathrm{O_2(g)} \rightarrow \mathrm{CO(g)} + \mathrm{SiO(g)} \quad \Delta G_r = -460.8 \; kJ/mol \tag{3}$$

$$\mathrm{SiC(s)} + 1/2\mathrm{O_2}\,(g) \rightarrow \mathrm{C(s)} + \mathrm{SiO(g)} \quad \Delta G_r = -191.6 \; kJ/mol \tag{4}$$

$$\mathrm{SiO(g)} + 1/2\mathrm{O_2}\,(g) \rightarrow \mathrm{SiO_2(s)} \quad \Delta G_r = -514.8 \; kJ/mol \tag{5}$$

where (s) and (g) are solid and gaseous phases, respectively.

All reactions are characterized by significantly negative free Gibbs energy ($\Delta G_r$). The $\Delta G_r$ values were calculated using reliable data [42] taking into account the dependences of entropies, enthalpies and heat capacities of all the reactants and products on temperature. Despite that reaction (3) has a more negative $\Delta G_r$ value than reaction (4), no Si is present in the EDX data of all samples studied (see Supplementary Materials). One can suggest that reaction (5) resulting in gaseous SiO and solid C is more favorable from the point of view of kinetics. The resulted SiO is removed from the ceramic matrix, readily transforms into SiO$_2$ according to reaction (6) and stays in the powder bed. Thus, the reaction (4) and (5) sequence takes place during sintering.

Thus, it can be concluded that the diffusion of SiC from a powder bed to the ceramic matrix followed by its decomposition takes place during annealing at 1550 °C. Its effects on the microstructure and electrical properties of the sintered samples depend on the initial rGO concentration in the composite powder. In summary, the suggested powder bed sintering technique in a three-layer graphite/SiC/graphite powder bed allows one to obtain well-formed dense YSZ-rGO ceramics and can become a suitable alternative to existing equilibrium (conventional sintering) and non-equilibrium (SPS, flash sintering) methods for various oxide ceramic composite fabrication.

## 5. Conclusions

Using SEM, XRD, electrochemical impedance spectroscopy and mechanical tests, it was shown that the increase of rGO content up to 1 wt.% in the sintered composite led to the microstructure improvement, density close to the theoretical one, high crystallinity and microhardness up to 13 GPa. Via the impedance spectroscopy, it was shown that Z0 possesses the highest conductivity values at temperatures >450 °C, while Z1 composite has the lowest conductivity among all the compositions and the highest activation energy of 1.35 eV. Using the impedance spectroscopy and EDX data, it was revealed that carbon phase agglomeration takes place in Z2 composite resulting in the damage of the 3D carbon subphase network. The performed thermodynamic analysis showed that SiC decomposition with the formation of free carbon is favorable and results in the formation of a porous near-edge layer for all investigated composites.

**Supplementary Materials:** The following supporting information can be downloaded at: https://www.mdpi.com/article/10.3390/app12010190/s1, Figure S1: SEM image of the cross-section of Z0 sample with pointed areas where EDX spectra were collected., Table S1. Distribution of elements in Z0 sample according to taken EDX spectra (in atomic %)., Figure S2. EDX spectrum of cross section of Z0 sample., Figure S3. SEM image of the cross-section of sample Z2 with pointed areas where EDX spectra were collected., Table S2. Distribution of elements in sample Z2 according to taken EDX spectra (in atomic %)., Figure S4. SEM image of Z0 ceramics bulk and carbon, and zirconium elemental distribution maps., Figure S5. Particle size distribution of used in powder bed SiC measured by dynamic light scattering method.

**Author Contributions:** Conceptualization, A.G. and V.K.; methodology, A.G. and V.K.; validation, O.G. and I.S.; formal anal E.B. and A.G.; investigation, I.S., O.G., E.B. and A.G.; resources, V.K. and O.K.; data curation, A.G. and O.K.; writing—original draft preparation, A.G.; writing—review and editing, A.G. and O.K; visualization, A.G.; supervision, V.K.; project administration, V.K.; funding acquisition, V.K. All authors have read and agreed to the published version of the manuscript.

**Funding:** This research was funded by Russian Science Foundation (RSF) grant number 18-19-00255.

**Institutional Review Board Statement:** Not applicable.

**Informed Consent Statement:** Not applicable.

**Data Availability Statement:** The data presented in this study are available on request from the corresponding author.

**Acknowledgments:** SEM and EDX measurements were performed at the Research Park of St. Petersburg State University Interdisciplinary Resource Centre for Nanotechnology. Raman spectroscopy data of sintered ceramics were obtained at the Centre for Optical and Laser Materials Research of St. Petersburg State University Research Park.

**Conflicts of Interest:** The authors declare that they have no known competing financial interests or personal relationships that could have appeared to influence the work reported in this paper.

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
