# Peer review of "Phase Formation and the Electrical Properties of YSZ/rGO Composite Ceramics Sintered Using Silicon Carbide Powder Bed"

_applsci, doi:10.3390/app12010190_

Round 1
Reviewer 1 Report
Line 19: “The effects”
Line 38: “The fabrication of stabilized zirconia/graphene composites requires a high temperature sintering (1400-1600 ºC)”. Please, change and put: “(1300-1600 ºC)”. There are some authors like [1][2][3] who sintered zirconia/graphene composites using a temperature of 1300 ºC, please mention these authors.
Line 44: It is better if you remove the reference [14]. It is most appropriate if you put a Review of this topic, about sintering methods of reinforced graphene zirconia composites. You can put the references [4] [5].
Line 140: “previous works of authors [8,9]”.
Line 144: “with a force of 5t”. Please put in the International System of Units, use newtons.
Line 147: “at 1550 ºC for 3 hours in air”. Why did you employ these values? Please put references.
Section 2.2. Characterization of obtained YSZ-rGO ceramics. You should include in this section the etching of the samples. Thermal etching (temperature/s, time/s, in air or other atmosphere…) or chemical etching (solution, concentration of components, time inside the solution…). You put SEM micrographs with grain boundaries but you need to do etching to the samples to watch them.
Line 189: Equation (2) is the equation (1).
Line 199: “of the sintered Z0-Z2”.
Line 227/233: You put the reference [28] in the line 227 and you pass directly in the line 233 to the reference [32]. I think that the reference [32] in the text should be the reference [29], please check it and put order in all the references in the text and also in the list of references. Other thing in the line 233 concerning to the reference [32], is really this reference in the list of references? I think it must be the reference [34], [35] or [36] in the final list of references because these three references (34, 35 and 36) speak about Raman spectroscopy of SiC and Si.
Line 252: “Along with that, they contain a number of pores and their amount increases with the increases with the increase of rGO content”. Why? Reason?
Line 259: “contain a number of prolongated cracks…, their amount significantly decreases with the addition of rGO content to…”. Please mention that the fracture toughness increases due to the addition of rGO which stops the propagation of cracks in the ceramic matrix.
Line 276: “Figure 7”, use the same font.
Line 304: It is the equation (1), not the equation (2).
Line 396: It is the equation (2).
Line 397: It is the equation (3).
Line 399: It is the equation (4).
Line 401: “…(2), equilibrium (4)…”
Line 405: It is the equation (5).
Line 406: It is the equation (6).
Line 407: It is the equation (7).
Line 408: It is the equation (8).
Line 409: It is the equation (9).
Line 410: “by the reaction (5)…”
Line 414: “…so the reactions (6) and (8) are dominant, especially (6)…”
- Cano-Crespo, R.; Rivero-Antúnez, P.; Gómez-García, D.; Moreno, R.; Domínguez-Rodríguez, A. The possible detriment of oxygen in creep of alumina and zirconia ceramic composites reinforced with graphene. Materials (Basel). 2021, 14, doi:10.3390/ma14040984.
- Rodríguez-Rojas, F.; Cano-Crespo, R.; Borrero-López, O.; Domínguez-Rodríguez, A.; Ortiz, A.L. Effect of 1-D and 2-D carbon-based nano-reinforcements on the dry sliding-wear behaviour of 3Y-TZP ceramics. J. Eur. Ceram. Soc. 2021, 41, doi:10.1016/j.jeurceramsoc.2020.12.054.
- Porwal, H.; Grasso, S.; Reece, M.J. Review of graphene-ceramic matrix composites. Adv. Appl. Ceram. 2013, 112, 443–454, doi:10.1179/174367613X13764308970581.
- Glukharev, A.G.; Konakov, V.G. Synthesis and properties of zirconia-graphene composite ceramics: A brief review. Rev. Adv. Mater. Sci. 2018, 56, 124–138, doi:10.1515/RAMS-2018-0041.
Author Response
Dear reviewer!
Thank you for a prompt review, your time and valuable comments on the manuscript. All of them were taken into the account in the revised manuscript. The changes are highlighted in yellow. Besides the manuscript has been grammar, and spell checked, and corrected by a colleague, who is a native speaker. Let us answer questions and comments one by one in details
Reviewer 1:
Line 19: “The effects”
Line 38: “The fabrication of stabilized zirconia/graphene composites requires a high temperature sintering (1400-1600 ºC)”. Please, change and put: “(1300-1600 ºC)”. There are some authors like [1][2][3] who sintered zirconia/graphene composites using a temperature of 1300 ºC, please mention these authors.
The specified articles have been added as references
Line 44: It is better if you remove the reference [14]. It is most appropriate if you put a Review of this topic, about sintering methods of reinforced graphene zirconia composites. You can put the references [4] [5].
Reference [14] has been removed and replaced with review reference [4]
Line 140: “previous works of authors [8,9]”.
Line 144: “with a force of 5t”. Please put in the International System of Units, use newtons. Used pressing force was recalculated to newtons.
Line 147: “at 1550 ºC for 3 hours in air”. Why did you employ these values? Please put references.
Chosen condition were the same as for sintering of YSZ-rGO system without powder bed. Corresponding reference has been added
Section 2.2. Characterization of obtained YSZ-rGO ceramics. You should include in this section the etching of the samples. Thermal etching (temperature/s, time/s, in air or other atmosphere…) or chemical etching (solution, concentration of components, time inside the solution…). You put SEM micrographs with grain boundaries but you need to do etching to the samples to watch them.
No etching procedure was used. Grain boundaries on the SEM images were visible for as-received samples after polishing. The corresponding mention has been given in the text.
Line 189: Equation (2) is the equation (1). Scherrer’s formula for dcryst calculation was added as the equation (1)
Line 199: “of the sintered Z0-Z2”.
Line 227/233: You put the reference [28] in the line 227 and you pass directly in the line 233 to the reference [32]. I think that the reference [32] in the text should be the reference [29], please check it and put order in all the references in the text and also in the list of references. Other thing in the line 233 concerning to the reference [32], is really this reference in the list of references? I think it must be the reference [34], [35] or [36] in the final list of references because these three references (34, 35 and 36) speak about Raman spectroscopy of SiC and Si. This mistake was fixed, the correct reference has been added
Line 252: “Along with that, they contain a number of pores and their amount increases with the increases with the increase of rGO content”. Why? Reason? The most probable reason for that is initial rGO agglomeration process in the near-surface layer. The corresponding mention has been given in the text.
Line 259: “contain a number of prolongated cracks…, their amount significantly decreases with the addition of rGO content to…”. Please mention that the fracture toughness increases due to the addition of rGO which stops the propagation of cracks in the ceramic matrix. This moment has been mentioned in the text.
Line 276: “Figure 7”, use the same font.
Line 304: It is the equation (1), not the equation (2).
Line 396: It is the equation (2).
Line 397: It is the equation (3).
Line 399: It is the equation (4).
Line 401: “…(2), equilibrium (4)…”
Line 405: It is the equation (5).
Line 406: It is the equation (6).
Line 407: It is the equation (7).
Line 408: It is the equation (8).
Line 409: It is the equation (9).
Line 410: “by the reaction (5)…”
Line 414: “…so the reactions (6) and (8) are dominant, especially (6)…”
Text was double-checked and all mentioned mistakes have been fixed.
- Cano-Crespo, R.; Rivero-Antúnez, P.; Gómez-García, D.; Moreno, R.; Domínguez-Rodríguez, A. The possible detriment of oxygen in creep of alumina and zirconia ceramic composites reinforced with graphene. Materials (Basel). 2021, 14, doi:10.3390/ma14040984.
- Rodríguez-Rojas, F.; Cano-Crespo, R.; Borrero-López, O.; Domínguez-Rodríguez, A.; Ortiz, A.L. Effect of 1-D and 2-D carbon-based nano-reinforcements on the dry sliding-wear behaviour of 3Y-TZP ceramics. J. Eur. Ceram. Soc. 2021, 41, doi:10.1016/j.jeurceramsoc.2020.12.054.
- Porwal, H.; Grasso, S.; Reece, M.J. Review of graphene-ceramic matrix composites. Adv. Appl. Ceram. 2013, 112, 443–454, doi:10.1179/174367613X13764308970581.
- Glukharev, A.G.; Konakov, V.G. Synthesis and properties of zirconia-graphene composite ceramics: A brief review. Rev. Adv. Mater. Sci. 2018, 56, 124–138, doi:10.1515/RAMS-2018-0041.
Reviewer 2 Report
The manuscript investigated the sintering technique of fully stabilized zirconia composites modified with grapheme in the SiC powder bed. In addition, the properties of specimens are characterizated. It is interesting and novel and will work well in the optimization and development of ceramic matrix composites. However, there were some scientific questions should be revised:
- What are the advantages of the atmosphere condition over other atmospheres or vacuum sintering?
- In Figure 2 and Figure 6, it could be seen that the obvious cracks in Z1, does the cracks influence the properties of the samples?
- The testing method of crystallinity and crystallite size should be given.
- The formation mechanism of the cracks in Figure. 5 and Figure. 6 should be explained in detail.
- In the section of “Discussion”, I am confused about the decomposition of silicon carbide. From the references of 37-41, I could not find the equation (5), (6) and (7) in your manuscript. And I read the references of 37-41, the reaction conditions of the equation (5), (6) and (7) were different to the experiment in this manuscript. I calculated the Gibbs free energy of the equation of (5), (6) and (7) by FactSage, the reactions could not occurred. In my opinion, the reaction mechanism should be re-discuss.
Author Response
Dear reviewer!
Thank you for a prompt review, your time and valuable comments on the manuscript. All of them were taken into the account in the revised manuscript. The changes are highlighted in yellow. Besides the manuscript has been grammar, and spell checked, and corrected by a colleague, who is a native speaker. Let us answer questions and comments one by one in details.
The manuscript investigated the sintering technique of fully stabilized zirconia composites modified with grapheme in the SiC powder bed. In addition, the properties of specimens are characterizated. It is interesting and novel and will work well in the optimization and development of ceramic matrix composites. However, there were some scientific questions should be revised:
- What are the advantages of the atmosphere condition over other atmospheres or vacuum sintering?
Use of argon or deep vacuum for the successful sintering requires a significant complication of the experimental setup (high-power vacuum pumps, oxygen control sensors, etc.), which increases the cost and imposes restrictions on the number of simultaneously sintered samples. Sintering on air requires only furnace. It was mentioned in the text (line 193).
- In Figure 2 and Figure 6, it could be seen that the obvious cracks in Z1, does the cracks influence the properties of the samples?
Central sectors (not containing cracks) were cut from each specimen for the electrochemical investigations and mechanical testing. Presented in fig. 2 samples with cracks were obtained after tensile strength testing, which was unsuccessful.
The testing method of crystallinity and crystallite size should be given.
Sherrer’s formula for dcryst calculation was added as the equation (1)
- The formation mechanism of the cracks in Figure. 5 and Figure. 6 should be explained in detail.
Pore formation mechanism was discussed by the means of different CTE for SiC and YSZ. (lines 413-418)
- In the section of “Discussion”, I am confused about the decomposition of silicon carbide. From the references of 37-41, I could not find the equation (5), (6) and (7) in your manuscript. And I read the references of 37-41, the reaction conditions of the equation (5), (6) and (7) were different to the experiment in this manuscript. I calculated the Gibbs free energy of the equation of (5), (6) and (7) by FactSage, the reactions could not occurred. In my opinion, the reaction mechanism should be re-discuss.
Silicon carbide decomposition mechanism was rewritten with the calculation of Gibbs free energy.
Reviewer 3 Report
In this article, the authors describe the development of a novel powder bed sintering method for the production of ceramics/graphene composites. Characterization using SEM, Raman, and XRD as wall as electrochemical impedance for conductivity measurements were used to describe the resulting composites. Mechanical properties were also explored. The resulting reduced graphene oxide exhibited enhanced strength and conductivity. While the science is of high quality, the article should go through another round of proofreading for syntax, grammar, and style. Publication is recommended after minor revisions.
Below are just some suggested minor revisions at the following lines.
-19. Remove "The no"
-22. Change to "... allowed well-formed dense YSZ/rGO ceramics to be obtained and..."
-22. "... become a suitable ..."
-33. remove comma.
-40. Change "has" to "have"
-61. Change to "techniques"
-99. "compared to"
-255. a,b,c,d should be described in the caption.
-258. decreases
-282. Perhaps consider placing figure 8 in the suppinfo with text referencing it in the manuscript.
-337. remove extra period.
-391. Let us?
-394. change throw to through.
-396-409. It is not clear why different reaction arrows are used; forward vs both directions.
-428. "lead" should be "led"
Author Response
Dear reviewer!
Thank you for a prompt review, your time and valuable comments on the manuscript. All of them were taken into the account in the revised manuscript. The changes are highlighted in yellow. Besides the manuscript has been grammar, and spell checked, and corrected by a colleague, who is a native speaker. Let us answer questions and comments one by one in details.
In this article, the authors describe the development of a novel powder bed sintering method for the production of ceramics/graphene composites. Characterization using SEM, Raman, and XRD as wall as electrochemical impedance for conductivity measurements were used to describe the resulting composites. Mechanical properties were also explored. The resulting reduced graphene oxide exhibited enhanced strength and conductivity. While the science is of high quality, the article should go through another round of proofreading for syntax, grammar, and style. Publication is recommended after minor revisions.
Below are just some suggested minor revisions at the following lines.
-19. Remove "The no"
-22. Change to "... allowed well-formed dense YSZ/rGO ceramics to be obtained and..."
-22. "... become a suitable ..."
-33. remove comma.
-40. Change "has" to "have"
-61. Change to "techniques"
-99. "compared to"
-255. a,b,c,d should be described in the caption.
-258. decreases
-282. Perhaps consider placing figure 8 in the suppinfo with text referencing it in the manuscript.
Fig. 8 has been moved to supplementary materials as figure S4. The corresponding mention has been given in the text.
-337. remove extra period.
-391. Let us?
-394. change throw to through.
-396-409. It is not clear why different reaction arrows are used; forward vs both directions.
Silicon carbide decomposition has been rediscussed with new mechanism.
-428. "lead" should be "led"
Text was double-checked and all mentioned mistakes have been fixed.
Round 2
Reviewer 1 Report
The paper has improved and it can be accepted in the present form.
Reviewer 2 Report
Most of my previous comments being without any reply.